# Incidence of Post-Traumatic Stress Disorder after Coronavirus Disease

**DOI:** 10.3390/healthcare8040373

**Published:** 2020-09-30

**Authors:** Min Cheol Chang, Donghwi Park

**Affiliations:** 1Department of Physical Medicine and Rehabilitation, College of Medicine, Yeungnam University, Daegu 38541, Korea; wheel633@ynu.ac.kr; 2Department of Physical Medicine and Rehabilitation, Ulsan University Hospital, University of Ulsan College of Medicine, Ulsan 44033, Korea

**Keywords:** post-traumatic stress disorder, coronavirus disease, infection, pandemic, mental health

## Abstract

*Background*: The coronavirus disease (COVID-19) emerged from China and rapidly spread to many other countries. In this study, we investigated the prevalence of post-traumatic stress disorder (PTSD) among patients with COVID-19 who were treated and discharged from a university hospital in Daegu, Korea. *Methods*: In total, 64 patients who were diagnosed with COVID-19 and then hospitalized, treated and discharged from the university hospital between February and April 2020 participated in our study. We conducted telephone interviews with the participants and evaluated the presence of PTSD using the Post-Traumatic Stress Disorder Checklist-5 (PCL-5) based on the Diagnostic and Statistical Manual of Mental Disorders (DSM-5; score range: 0–80). If a score of ≥33 was obtained, then a diagnosis of PTSD was made. We analyzed the association between PTSD and demographic and clinical characteristics using the Mann–Whitney U and chi-square tests. *Results*: In total, 13 patients had a PCL-5 score of ≥33, which indicated that 20.3% (*n* = 64) of the patients had PTSD. No significant differences were observed in demographic characteristics, including, sex, age, hospitalization time and duration after discharge, between patients with PTSD and those without PTSD. *Conclusions*: The prevalence rate of PTSD was 20.3% in patients with COVID-19 who had been hospitalized, treated and discharged. Accordingly, clinicians should be aware of the high possibility of PTSD among COVID-19 patients. Mental health interventions supporting the mental health of patients should be provided to affected patients.

## 1. Introduction

The coronavirus disease (COVID-19) outbreak, which started in Wuhan, China, in December 2019, reached pandemic status by March 2020 [1,2,3,4]. On March 11, 2020, the World Health Organization (WHO) declared the COVID-19 outbreak a pandemic [5]. In Daegu, Korea, a massive outbreak of COVID-19 occurred between February and April 2020 [6]. Of the total population in Daegu, which is approximately 2.5 million, nearly 6200 people were infected with severe acute respiratory syndrome coronavirus 2 (SARS-CoV-2) during this period [6,7]. 

Approximately 2–5% of COVID-19 patients die due to progressive respiratory failure and massive alveolar damage [8]. The highly contagious nature of SARS-CoV-2 has been a major reason for the increasing number of deaths due to COVID-19 in almost all countries, although the mortality rate of COVID-19 is lower than that experienced for severe respiratory syndrome (SARS) and Middle East respiratory syndrome (MERS) [9]. All affected countries are actively taking measures to prevent the spread of COVID-19 and to decrease mortality. Many countries have employed strategies to prevent the secondary and tertiary transmission of COVID-19 by identifying, testing and isolating patients with the disease. However, as of August 2020, vaccines and/or curative approaches have not been used in real clinical settings, although a few vaccines and/or treatment methods have been tested in clinical trials. 

To date, several studies have reported risk factors for COVID-19’s progression, such as age, hypertension, use of angiotensin-converting enzyme inhibitors, and neurological disorders [8,10,11,12,13]. For example, young individuals infected with SARS-CoV-2 generally show milder respiratory symptoms than older adults [13]. Additionally, older adults (aged 60 years or older) with COVID-19 are likely to show massive alveolar damage and progressive respiratory failure, leading to a mortality rate of >10%. The mortality rate drastically increases with age in people aged 60 years and older [14,15]. Moreover, patients with Alzheimer’s dementia and COVID-19 have higher mortality rates than patients with only COVID-19 [8]. 

Knowledge of the risk factors for COVID-19 mortality can affect a person’s perception of the likelihood of death from COVID-19, not only in COVID-19 patients with risk factors but also in COVID-19 patients without risk factors. This may lead to a traumatizing experience and could cause psychiatric symptoms in patients with COVID-19. Post-traumatic stress disorder (PTSD) is a stress-related psychological illness that occurs immediately after a trauma, such as a severe accident or exposure to violence [16]. The emotions, memories and thoughts experienced during the trauma recur in the patient, leading to inconveniences and restrictions in their daily lives [16,17]. This is because approximately 2–5% of COVID-19 cases result in death due to massive alveolar damage and progressive respiratory failure [15]. Although there is some debate over the correlation between COVID-19 and PTSD, PTSD may still occur in patients infected with SARS-CoV-2. In previous disease outbreaks, the prevalence of PTSD after developing a serious infectious disease ranged from 4% to 41% in the general population [18]. For the COVID-19 outbreak, Cindy et al. reported that the prevalence of PTSD in young adults aged 18–30 years in the United States (US) was 31.8% [19]. Additionally, Luna et al. reported that the prevalence of PTSD-related symptoms was approximately 5% in Wuhan, the first region in China to be affected by the COVID-19 pandemic [20]. Only a few studies have addressed this important issue, although a high prevalence of PTSD is expected among patients infected with SARS-CoV-2 [19,21,22]. Some factors, such as interpersonal conflict, lower socioeconomic status, female sex, frequent use of social media and lower resilience and social support, have been reported to increase the risk of PTSD [23]. In this study, we aimed to determine the prevalence of PTSD among patients with COVID-19 who were treated in a university hospital in Daegu, Korea.

## 2. Methods

### 2.1. Patients

The study was approved by the Institutional Review Board (IRB) of Yeungnam University Hospital (2020-04-089). We recruited 107 patients (*n* = 107) who were first diagnosed with COVID-19 and then hospitalized, treated and discharged from a university hospital in Daegu, Korea, between February and April 2020. In our study, all adult patients who were diagnosed with COVID-19 according to the WHO interim guidance were screened [24]. 

### 2.2. Laboratory Procedures

Before 31 January 2020, a pan-coronavirus reverse-transcription polymerase chain reaction (RT-PCR) test was used to detect SARS-CoV-2 in Korea. The pan-coronavirus RT-PCR assay first analyzes the suspected clinical sample for all the coronaviruses [25]. If a positive reaction is detected in the test, a second test is performed using gene sequencing to determine whether the coronavirus is SARS-CoV-2 [25]. This test requires two separate assays over a 24-h testing period [26].

After 31 January 2020, all local government Public Health and Environmental Research Institutes started to diagnose COVID-19 using the RT-PCR kit approved by the Korean Centers for Disease Control and Prevention (KCDC) and the Korean Ministry of Food and Drug Safety [25]. Since 7 February 2020, more than 50 private medical institutions in Korea, which were approved by the KCDC, have started to use the RT-PCR kit to detect the presence of SARS-CoV-2 [25]. Since rRT-PCR-based assays usually detect only 2–3 of these genes, the assay allows for rapid testing and diagnosis [25].

Both nasopharyngeal and oropharyngeal swabs were used to detect SARS-CoV-2. Therefore, in all patients with COVID-19, the diagnosis was confirmed through RT-PCR (Allplex^TM^ 2019-nCoV Assay, Seegene, South Korea) using nasopharyngeal and oropharyngeal swabs obtained at the hospital. 

### 2.3. Treatment Methods

The patients were treated with either hydroxychloroquine sulfate (Oxiklorine^®^, Myungmoon Pharm. Co., Ltd., Seoul, South Korea; 400 mg qd per day) only or a combination of hydroxychloroquine sulfate (Oxiklorine^®^, Myungmoon Pharm. Co., Ltd., Seoul, South Korea; 400 mg) and lopinavir/ritonavir (400 mg/100 mg bid per day; Kaletra^®^, AbbVie Inc., North Chicago, IL, USA) [10]. After symptom improvement, the patients were discharged when they met the following criteria: no fever without the administration of antipyretics and negative results on PCR tests performed twice in a 24-h interval. 

### 2.4. Evaluation of PTSD

The presence of PTSD was investigated through telephone interviews. Of the 107 patients, 43 had missing contact information, did not answer their phones, or refused to participate. The remaining 64 patients completed a telephone interview. The demographic data of these patients were extracted through chart reviews completed during the interview, and information on sex, age, hospitalization time and post-discharge duration was collected.

### 2.5. PCL-5

To evaluate the presence of PTSD, the Post-traumatic Stress Disorder Checklist-5 (PCL-5), developed to correspond to the criteria of the Diagnostic and Statistical Manual of Mental Disorders (DSM-5), was used [27]. The PCL-5 is a 20-item self-report tool involving a five-point Likert-type scale, with scores ranging from “Not at all” (0) to “Extremely” (4), resulting in a symptom severity score of between 0 and 80; the PCL-5 assesses the presence and severity of PTSD symptoms. For the current study, if a score of ≥33 was obtained, then a diagnosis of PTSD was made [20].

### 2.6. Statistical Analysis

The Kolmogorov–Smirnov test was used to determine whether the data conformed to a normal distribution. We analyzed the association between the presence of PTSD and demographic and clinical characteristics using the Mann–Whitney U and chi-square tests. The acceptable statistical significance was set at a *p*-value of <0.05. All statistical analyses were conducted using IBM SPSS Statistics for Windows, version 23.0 (IBM Corp., Armonk, NY, USA).

## 3. Results

### 3.1. Characteristics of Patients with COVID-19

Table 1 shows the baseline characteristics of all COVID-19 patients and a comparison of these characteristics between the PTSD and non-PTSD groups. Of the 64 patients who participated in the study, 13 had a PCL-5 score of ≥33, which indicated that the prevalence of PTSD was 20.3% (*n* = 64) in the patients. The mean patient age was 54.7 ± 16.6 years, and the sex ratio was 28:36 (male:female). The mean PCL-5 score was 17.0 ± 17.1, the mean duration of admission was 31.2 ± 18.1 days, and the mean duration after discharge was 75.7 ± 20.0 days (Table 1).

### 3.2. Differences in the Demographic Characteristics of COVID-19 Patients in the PTSD and Non-PTSD Groups

The mean PCL-5 scores in the PTSD and non-PTSD groups were 46.0 ± 11.9 and 9.6 ± 7.6, respectively, and there was a significant difference between the PTSD and non-PTSD groups (*p* < 0.001; Table 1). The sex ratios (male:female) in the PTSD and non-PTSD groups were 4:9 and 24:27, respectively. Although the proportion of men was higher in the PTSD group than in the non-PTSD group, there was no significant difference (*p* > 0.05; Table 1).

The mean ages in the PTSD and non-PTSD groups were 57.8 ± 15.2 and 53.9 ± 17.0 years, respectively (Table 1). Although the mean patient age in the PTSD group was higher than that of the non-PTSD group, there was no significant difference (*p* > 0.05). The mean durations of admission in the PTSD and non-PTSD groups were 35.7 ± 21.3 and 30.1 ± 17.2 days, respectively (Table 1). Although the mean duration of admission was higher in the PTSD group than in the non-PTSD group, the difference was not significant (*p* > 0.05). The mean durations from discharge in the PTSD and non-PTSD groups were 69.8 ± 25.4 and 77.2 ± 18.4 days, respectively (Table 1). Although the mean duration from discharge in the PTSD group was shorter than that of the non-PTSD group, there was no significant difference (*p* > 0.05).

Therefore, there were no significant differences in demographic characteristics (sex, age, hospitalization time and duration after discharge) between patients with PTSD and those without PTSD.

## 4. Discussion

Although the death rate associated with COVID-19 is relatively lower than that of SARS and MERS due to the extremely highly contagious nature of SARS-CoV-2, the number of cases of death from COVID-19 has being rapidly increasing in nearly all counties. In our study, approximately 20% of patients infected with SARS-CoV-2 were admitted to hospital. However, sex, age, duration of admission and the interval between discharge and the interview were not associated with the development of PTSD.

Park et al., who conducted a study on 63 of the 148 patients in Korea who also developed MERS and survived, reported that PTSD occurred in 42.9% of the patients one year after full recovery [28]. Regardless of the severity of the infection, the PTSD risk was higher in cases in which the survivors had perceptions of high levels of social stigma against patients with infection or in which the survivors had increased anxiety levels [28]. Another report showed that 42% of Chinese patients who were infected with SARS had PTSD even after four years [29]. The incidence of PTSD after the development of COVID-19 (20%) was lower than that after the development of MERS or SARS [29]. Although this could be explained by differences in the duration between the outbreak and the investigation of the presence of PTSD, it is also possible that the lower mortality rate (2–5%) in patients with COVID-19 could have affected the incidence of PTSD in COVID-19 patients [30].

Regarding other psychological disorders in patients with COVID-19, Ettman et al. recruited 1441 participants and found that 27.8% had depression symptoms, compared with 8.5% before COVID-19 [31]. Further, Salari et al. performed a meta-analysis to evaluate the prevalence of various psychological disorders during the COVID-19 pandemic, and reported that the prevalence of stress, anxiety and depression was 29.6%, 31.9% and 33.7, respectively [32]. 

Several studies are currently being performed on the incidence of PTSD during the COVID-19 pandemic [7,8,9,10]. In April 2020, Fekih-Romdhane et al. [21] investigated the prevalence of PTSD in 603 Tunisian people, and reported that 33% of the participants had PTSD. Liu et al. [19] found similar results (31.8%) while evaluating the incidence of PTSD in 898 young American adults between 13 April 2020, and 19 May 2020. Forte et al. [22] investigated the presence of PTSD in 2286 Italians during a massive COVID-19 outbreak in Italy and found that PTSD was prevalent in 29.5% of the participants. Lastly, Wang et al. [33] evaluated 202 nurses exposed to COVID-19 patients in Hubei, China, and reported that 16.8% of them had PTSD. However, these studies included both individuals who were not diagnosed with COVID-19 and those who were diagnosed with COVID-19. In contrast, our study exclusively included patients who were diagnosed with COVID-19 and were hospitalized, treated, and discharged. 

In the present study, PTSD was prevalent in 20.3% of patients with COVID-19 who were discharged after a full recovery following treatment. PTSD restricts patients from living their lives normally, and precipitates mental disorders such as depression, schizophrenia and alcohol addiction. Thus, clinicians should be aware of the possibility of PTSD among COVID-19 patients and should provide appropriate treatment to individuals who have relevant symptoms. In particular, COVID-19 patients are at a moderate risk of PTSD, especially those who have been discharged after a full recovery; thus, PTSD detection through active inquiry and appropriate subsequent treatment is warranted.

The current study has a few limitations. First, the sample size was small. In the future, studies with a larger number of COVID-19 patients will be necessary in order to discover the risk factors for PTSD in this population. Second, other psychiatric symptoms, including depression, anxiety and sleep disorders, were not evaluated. Although the PCL-5 is a useful tool for evaluating PTSD, other clinical tools that can evaluate other psychiatric symptoms, such as depression and anxiety, were not used in this study. Third, PTSD might not be the most appropriate diagnosis because the PCL-5 is usually used for evaluating the degree of post-traumatic stress symptoms, not confirming a diagnosis of PTSD. Therefore, further studies involving the use of various psychological evaluation tools are necessary in order to investigate the mental health status of patients with COVID-19. Finally, several variables that could be potential risk factors for PTSD, such as a history of mental illness, personal characteristics and social support, were not investigated in the present study. Therefore, further studies are needed in order to overcome these limitations.

## Figures and Tables

**Table 1 healthcare-08-00373-t001:** Characteristics of the study participants according to the presence of post-traumatic stress disorder.

	PTSD Group	Non-PTSD Group	*p*-Value	Total
Number of patients (n)	13	51		64
PCL-5 score	46.0 ± 11.9	9.6 ± 7.6	<0.001	17.0 ± 17.1
M:F (n)	4:9	24:27	0.291	28:36
Age (years)	57.8 ± 15.2	53.9 ± 17.0	0.526	54.7 ± 16.6
Admission duration (days)	35.7 ± 21.3	30.1 ± 17.2	0.499	31.2 ± 18.1
Duration after discharge (days)	69.8 ± 25.4	77.2 ± 18.4	0.457	75.7 ± 20.0

PTSD: post-traumatic stress disorder; M: male; F: female; PCL: Post-traumatic Stress Disorder Checklist; PTSD group: patients with PTSD; Non-PTSD group: patients without PTSD.

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
