# Peer review of "Incidence of Post-Traumatic Stress Disorder after Coronavirus Disease"

_healthcare, 2020, doi:10.3390/healthcare8040373_

Round 1
Reviewer 1 Report
This is an interesting article, though somewhat hampered by the sample size in its conclusions and several methodological issues.
Generally:
I start with this point, because it is important for the entire article. Your interpretation of what PTSD is, is wrong if you follow the DSM-V. Yes, the PCL-5 gives indications of the severity of PTSD, but you can only diagnose PTSD if you check for the criteria, as mentioned in the DSM-V, but also in the guide on the PCL-5. The PCL-5 has 20 questions and certain questions correspond to certain symptom clusters (which are represented in a criterium). What your score now mainly indicates, is that 13 patients had symptoms that indicate severe stress, or posttraumatic symptoms – but having symptoms is different from having the disorder. Thus, check also for the different criteria.
Furthermore, PTSD might not be the appropriate diagnosis. Perhaps change PTSD to PTSS, to indicate you are investigating the posttraumatic symptoms. It is difficult to state that you have diagnosed PTSD, because, from the information I find in this article, you have not, for several reasons. To mention a few: you did not assess criterion A (see further), you did not assess whether the stress is caused by another psychopathology, et cetera.
Another general point: please let someone who is a native speaker reread this. That is not to say the English is bad here – anything but that! – there are however at times sentences where the meaning of what you mean is lost. They are grammatically correct, but they do not seem to mean what you intended.
Abstract:
In methods, there is no mention of the methods used to analyse.
Introduction:
Line 58-59: I would be careful to state that PTSD is linked with anxiety. If you mean by association, yes, but I do not see why that would be important here. If you mean as psychological illness, then the DSM-V disagrees with you, because as of the DMS-V, PTSD is a stress-related psychological illness, not anxiety-related (which it was in the DSM-IV).
Line 62-63: It is not clear why the authors consider COVID-19 as life-threatening, as there is no source. First, there is an argument to be made here that COVID-19 is NOT something which can cause PTSD (in short: it is unclear how PTSD would cause a biological reaction of fight-flight-freeze, which is the biological cause of PTSD, which has to do with the amygdala, thalamus, et cetera). Several authors have suggested that 1, PTSD cannot be associated with COVID-19 (see for example the recent comment of Pfefferbaum and North in the New England Journal of Medicine). 2, Adjustment disorders are more appropriate as diagnosis. However, this is not to say you are wrong to study PTSD. In fact, I think there are arguments to be made why PTSD would occur with patients with COVID-19. However, if you state: COVID-19 can cause PTSD, you need to follow this immediately with several references that are about patient-populations.
Line 74-76: the PCL-5 is, to my knowledge, a scale developed to correspond to the criteria of the DSM-V, but not part of the DSM-V, which you seem to indicate. Thus, please indicate that it is developed to correspond to the criteria of the DSM-V.
Line 78-81: Everything concerning that the PCL 5 can study civilian populations or military populations is false and needs to be removed. You are referring here to the PCL-4, which indeed had a difference in scales for civilian and military. The PCL-5 does not have such a distinction. This can also be noticed in the year of the sources you cite here (which is before the publication of the DSM-5). This is important as PTSD is very different in some aspects in the DSM-5 when compared to the DSM-4.
Line 72-81: Some factors such as interpersonal conflicts, lower socioeconomic status, female sex, frequent 73 social media use, and lower resilience and social support have been reported to increase the risk of 74 PTSD.[23] The Post-traumatic Stress Disorder Checklist (PCL) from the Diagnostic and Statistical 75 Manual of Mental Disorders (DSM) is a widely used tool for evaluating self-reported PTSD 76 symptoms.[24] The PCL was recently revised as PCL-5 to reflect changes made to the PTSD criteria 77 in the DSM-5. PCL-5 item scores were summed to yield a continuous measure of PTSD symptom 78 severity.[24] Various cutoff scores, ranging from 30 to 60 points, for the PCL total severity score have 79 been used to diagnose PTSD depending on the population (e.g., civilian trauma vs. combat), setting 80 (e.g., primary care vs. PTSD clinic), and assessment goal (e.g., differential diagnosis vs. screening).[25, 81 26] Therefore, using PCL-5, we aimed to determine the prevalence of PTSD among patients with 82 COVID-19 who were treated at and discharged from a university hospital in Daegu, South Korea.
This should be shortened, as this is actually part of your methodology. So, I would suggest:
Some factors such as interpersonal conflicts, lower socioeconomic status, female sex, frequent social media use, and lower resilience and social support have been reported to increase the risk of PTSD.[23] In this study, we aimed to determine the prevalence of PTSD among patients with COVID-19 who were treated at and discharged from a university hospital in Daegu, South Korea.
Methodology:
Line 121: again, not from the DSM-5, if I’m not mistaken (which I might be).
Line 123: The most prominent changes in the PCL-5 included the following: (a) adding three items to evaluate three new symptoms of PTSD (negative emotions, blame, and reckless or self-destructive behavior), (b) rewording existing items to reflect changes in existing PTSD symptoms, and (c) changing the rating scale from 1–5 to 0–4 so that the lowest possible score is 0, which is more intuitive than 17 as on the PCL.
I do not understand 1, why this is mentioned here, 2, what it means that 17 is less intuitive? I think you might have misread what was mentioned in wherever you found this. The PCL had indeed 17 questions, but that has nothing to do with the rating scale.
I would just describe what is in the PCL-5 and leave the changes from the PCL to the PCL-5 out of this, because those changes are also based on the fact that the criteria have changed.
Statistical analysis: why use the independent t-test and the Mann-U Whitney? Does that mean that some of your analyses did not meet the assumptions? Perhaps mention that the Mann-u was used when the assumptions for the t-test were not met.
Evaluation of PTSD: who called these patients? Were these volunteers? Did they receive training in assessing PTSD?
Furthermore, you need to expand on how it is possible to recruit 107 people, and yet lose almost half of them. Did these people give permission for the interview beforehand? Or what is meant by recruitment?
Line 116-117: ‘provided additional information on PTSD’. What does this mean? I assume that this means you asked them the PCL-5 questions?
Important in this respect is if you assessed criterion A during this interview. What this means: when answering the PCL-5, did all the patients make clear they were referring to their COVID-19 experience? Because, if not, then the PTSD-rates could be caused by numerous things – for example, they might have been robbed before their infection.
Discussion:
Line 171: Although THE death rate
Line 174: were infected with SARS-CoV-2 and were admitted to the hospital due to PTSD.
They were admitted due to PTSD? No, COVID-19, right?
Furthermore, you state that in your study, 20% of the patients were infected… I thought everyone in your sample was infected?
You should probably just re-write this entire sentence. I suspect you mean that 20% of your sample had PTSD.
Line 181-182: SARS is not developed in people. They are, eg, infected with it. So, please re-write this sentence.
Line 197-198: alas, there are other studies on COVID-19 patients. Not many, I will agree, but there are…
Hosey M, et al.. Survivorship after COVID-19 ICU stay. Nat Rev Dis Primers. 2020; 6(1).
Guo Q, Zheng Y, Shi J, et al. Immediate psychological distress in quarantined patients with COVID-19 and its association with peripheral inflammation: A mixed-method study. Brain Behav Immun. 2020;88:17-27. doi:10.1016/j.bbi.2020.05.038
Krass P, et al. COVID-19 Outbreak Among Adolescents at an Inpatient Behavioral Health Hospital. J Adolesc Health. 2020 Aug 11;S1054-139X(20)30406-7.
Wesemann, et al. Influence of COVID-19 on general stress and posttraumatic stress symptoms among hospitalized high-risk patients. Psychol Med 2020 14;1-8.
Limitations:
Again, mention here that PTSD might not be the most appropriate diagnosis, cite Pfefferbaum and North for this or someone else.
Author Response
This is an interesting article, though somewhat hampered by the sample size in its conclusions and several methodological issues.
Generally:
I start with this point, because it is important for the entire article. Your interpretation of what PTSD is, is wrong if you follow the DSM-V. Yes, the PCL-5 gives indications of the severity of PTSD, but you can only diagnose PTSD if you check for the criteria, as mentioned in the DSM-V, but also in the guide on the PCL-5. The PCL-5 has 20 questions and certain questions correspond to certain symptom clusters (which are represented in a criterium). What your score now mainly indicates, is that 13 patients had symptoms that indicate severe stress, or posttraumatic symptoms – but having symptoms is different from having the disorder. Thus, check also for the different criteria.
Furthermore, PTSD might not be the appropriate diagnosis. Perhaps change PTSD to PTSS, to indicate you are investigating the posttraumatic symptoms. It is difficult to state that you have diagnosed PTSD, because, from the information I find in this article, you have not, for several reasons. To mention a few: you did not assess criterion A (see further), you did not assess whether the stress is caused by another psychopathology, et cetera.
Another general point: please let someone who is a native speaker reread this. That is not to say the English is bad here – anything but that! – there are however at times sentences where the meaning of what you mean is lost. They are grammatically correct, but they do not seem to mean what you intended.
Answer: I really appreciate your comments. I totally agree with your comments. However, complete change of PTSD into PTSS is not possible. Also, some studies suggested when PCL score of more than 32, it was considered as PTSD. Following your last comments on limitations, I think that comment that “PTSD might not be the most appropriate diagnosis because PCL-5 is usually used for evaluating the degree of post-traumatic stress symptoms, not confirming the diagnosis of PTSD”.
Also, by the native English speaker, our manuscript was edited again following your recommendation.
Abstract:
In methods, there is no mention of the methods used to analyse.
Answer: I appreciate your kind comment. We added the statistical analysis we used to the abstract.
Introduction:
Line 58-59: I would be careful to state that PTSD is linked with anxiety. If you mean by association, yes, but I do not see why that would be important here. If you mean as psychological illness, then the DSM-V disagrees with you, because as of the DMS-V, PTSD is a stress-related psychological illness, not anxiety-related (which it was in the DSM-IV).
Answer: We changed anxiety-related into a stress-related psychological illness.
Line 62-63: It is not clear why the authors consider COVID-19 as life-threatening, as there is no source. First, there is an argument to be made here that COVID-19 is NOT something which can cause PTSD (in short: it is unclear how PTSD would cause a biological reaction of fight-flight-freeze, which is the biological cause of PTSD, which has to do with the amygdala, thalamus, et cetera). Several authors have suggested that 1, PTSD cannot be associated with COVID-19 (see for example the recent comment of Pfefferbaum and North in the New England Journal of Medicine). 2, Adjustment disorders are more appropriate as diagnosis. However, this is not to say you are wrong to study PTSD. In fact, I think there are arguments to be made why PTSD would occur with patients with COVID-19. However, if you state: COVID-19 can cause PTSD, you need to follow this immediately with several references that are about patient-populations.
Answer: We think that COVID19 can be life-threatening condition because its mortality rate is 2-5% [ref1]. Also, I agree with your comment that there are some debates on relation between COVID19 and PTSS (or PTDS). We added this matter (debates) to the introduction section, and made the sentence related with this matter weaker.
[ref1] Kucharski AJ, Russell TW, Diamond C, Liu Y, Edmunds J, Funk S, et al. Early dynamics of transmission and control of COVID-19: a mathematical modelling study. Lancet Infect Dis. 2020;20(5):553-8. Epub 2020/03/15. doi: 10.1016/S1473-3099(20)30144-4.
Line 74-76: the PCL-5 is, to my knowledge, a scale developed to correspond to the criteria of the DSM-V, but not part of the DSM-V, which you seem to indicate. Thus, please indicate that it is developed to correspond to the criteria of the DSM-V.
Answer: I agree with your opinion. We indicated that PCL-5 was developed to correspond to the criteria of the DSM-V.
Line 78-81: Everything concerning that the PCL 5 can study civilian populations or military populations is false and needs to be removed. You are referring here to the PCL-4, which indeed had a difference in scales for civilian and military. The PCL-5 does not have such a distinction. This can also be noticed in the year of the sources you cite here (which is before the publication of the DSM-5). This is important as PTSD is very different in some aspects in the DSM-5 when compared to the DSM-4.
Answer: I appreciate your comments. Following your comments, we erased all the contents (… difference in scales for civilian and military…) related to your comments.
Line 72-81: Some factors such as interpersonal conflicts, lower socioeconomic status, female sex, frequent 73 social media use, and lower resilience and social support have been reported to increase the risk of 74 PTSD.[23] The Post-traumatic Stress Disorder Checklist (PCL) from the Diagnostic and Statistical 75 Manual of Mental Disorders (DSM) is a widely used tool for evaluating self-reported PTSD 76 symptoms.[24] The PCL was recently revised as PCL-5 to reflect changes made to the PTSD criteria 77 in the DSM-5. PCL-5 item scores were summed to yield a continuous measure of PTSD symptom 78 severity.[24] Various cutoff scores, ranging from 30 to 60 points, for the PCL total severity score have 79 been used to diagnose PTSD depending on the population (e.g., civilian trauma vs. combat), setting 80 (e.g., primary care vs. PTSD clinic), and assessment goal (e.g., differential diagnosis vs. screening).[25, 81 26] Therefore, using PCL-5, we aimed to determine the prevalence of PTSD among patients with 82 COVID-19 who were treated at and discharged from a university hospital in Daegu, South Korea.
This should be shortened, as this is actually part of your methodology. So, I would suggest:
Some factors such as interpersonal conflicts, lower socioeconomic status, female sex, frequent social media use, and lower resilience and social support have been reported to increase the risk of PTSD.[23] In this study, we aimed to determine the prevalence of PTSD among patients with COVID-19 who were treated at and discharged from a university hospital in Daegu, South Korea.
Answer: We changed above contents into those that the reviewer recommended.
Methodology:
Line 121: again, not from the DSM-5, if I’m not mistaken (which I might be).
Answer: We changed it into “developed to correspond to the criteria of the DSM-V”
Line 123: The most prominent changes in the PCL-5 included the following: (a) adding three items to evaluate three new symptoms of PTSD (negative emotions, blame, and reckless or self-destructive behavior), (b) rewording existing items to reflect changes in existing PTSD symptoms, and (c) changing the rating scale from 1–5 to 0–4 so that the lowest possible score is 0, which is more intuitive than 17 as on the PCL.
I do not understand 1, why this is mentioned here, 2, what it means that 17 is less intuitive? I think you might have misread what was mentioned in wherever you found this. The PCL had indeed 17 questions, but that has nothing to do with the rating scale.
I would just describe what is in the PCL-5 and leave the changes from the PCL to the PCL-5 out of this, because those changes are also based on the fact that the criteria have changed.
Answer: As the reviewer’s comments, above contents is not necessary, therefore we deleted them.
Statistical analysis: why use the independent t-test and the Mann-U Whitney? Does that mean that some of your analyses did not meet the assumptions? Perhaps mention that the Mann-u was used when the assumptions for the t-test were not met.
Answer: Because the number of PTSD group was only 13. Therefore, we used Mann-U Whitney test in this study. We corrected contents related to this matter.
Evaluation of PTSD: who called these patients? Were these volunteers? Did they receive training in assessing PTSD?
Furthermore, you need to expand on how it is possible to recruit 107 people, and yet lose almost half of them. Did these people give permission for the interview beforehand? Or what is meant by recruitment?
Answer: We recruited all the patients (n=117) who were hospitalized after diagnosis of COVID-19 and discharged. We get the verbal permission for the participation to the study by the phone. We corrected the related contents more clearly.
Line 116-117: ‘provided additional information on PTSD’. What does this mean? I assume that this means you asked them the PCL-5 questions?
Important in this respect is if you assessed criterion A during this interview. What this means: when answering the PCL-5, did all the patients make clear they were referring to their COVID-19 experience? Because, if not, then the PTSD-rates could be caused by numerous things – for example, they might have been robbed before their infection.
Answer: I think the sentence ‘provided additional information on PTSD’ is ambiguous. We deleted that.
Discussion:
Line 171: Although THE death rate
Answer: I appreciate your comment. We inserted “the” between although and death.
Line 174: were infected with SARS-CoV-2 and were admitted to the hospital due to PTSD.
They were admitted due to PTSD? No, COVID-19, right?
Furthermore, you state that in your study, 20% of the patients were infected… I thought everyone in your sample was infected?
You should probably just re-write this entire sentence. I suspect you mean that 20% of your sample had PTSD.
Answer: I appreciate your comments. Your comments are right. 20% of the patients diagnosed as COVID-19 had PTSD. We corrected sentence clearly.
Line 181-182: SARS is not developed in people. They are, eg, infected with it. So, please re-write this sentence.
Answer: I appreciate your comment. We corrected it into “infected with SARS”.
Line 197-198: alas, there are other studies on COVID-19 patients. Not many, I will agree, but there are…
Hosey M, et al.. Survivorship after COVID-19 ICU stay. Nat Rev Dis Primers. 2020; 6(1).
Guo Q, Zheng Y, Shi J, et al. Immediate psychological distress in quarantined patients with COVID-19 and its association with peripheral inflammation: A mixed-method study. Brain Behav Immun. 2020;88:17-27. doi:10.1016/j.bbi.2020.05.038
Krass P, et al. COVID-19 Outbreak Among Adolescents at an Inpatient Behavioral Health Hospital. J Adolesc Health. 2020 Aug 11;S1054-139X(20)30406-7.
Wesemann, et al. Influence of COVID-19 on general stress and posttraumatic stress symptoms among hospitalized high-risk patients. Psychol Med 2020 14;1-8.
Answer: I appreciate your comments. However, that study published in Nat Rev Dis Primers is not research paper (original study). It was the authors’ opinion on mental health (including PTSD) in patients with COVID-19. However, the other studies are related to PTSD or PTSS after COVID-19. This is recently published study. When we write our manuscript. There was no study on this topic. We deleted following sentence “this is the fist study on the prevalence of PTSD conducted exclusively among COVID-19 patient”
Limitations:
Again, mention here that PTSD might not be the most appropriate diagnosis, cite Pfefferbaum and North for this or someone else.
Answer: We added this matter to the discussion section as one of our limitations as follows:
Third, PTSD might not be the most appropriate diagnosis because PCL-5 is usually used for evaluating the degree of post-traumatic stress symptoms, not confirming the diagnosis of PTSD.
Reviewer 2 Report
The manuscript is really of interest in the field of COVID Research. However, I consider that the paper should be improved before being published. In the following lines, I will propose several minor changes to improve its quality.
Abstract. It should be clarified that the COVID-19 did not appear for a first time in Daegu, Korea. A brief mention about the Chinese outbreak should be noted.
In the methods section of the abstract it should be clarified whether the population was confined or not and whether there was any restriction in Korea in this period of time (February-April 2020).
Introduction.
Once again, a mention about the Chinese outbreak should be done. The authors introduced it but did not mention that it has been appeared in China.
The authors should also mention the prevalence of other psychiatric diagnoses associated with the experience of the illness by COVID (clinically relevant depressive symptoms, anxiety, psychotic symptoms, etc).
Methods. I have no comments. Psychometric instruments and all outcomes are well described.
Discussion. The authors should compare the prevalence of other psychiatric diagnoses in COVID-19 patients with the prevalence of PTSD.
Author Response
The manuscript is really of interest in the field of COVID Research. However, I consider that the paper should be improved before being published. In the following lines, I will propose several minor changes to improve its quality.
Abstract. It should be clarified that the COVID-19 did not appear for a first time in Daegu, Korea. A brief mention about the Chinese outbreak should be noted.
In the methods section of the abstract it should be clarified whether the population was confined or not and whether there was any restriction in Korea in this period of time (February-April 2020).
Answer: Following the reviewer’s comment, we added the contents on the Chinese outbreak. Also, we describe the recruited population in more detail by adding “Daegu, Republic of Korea”.
Introduction.
Once again, a mention about the Chinese outbreak should be done. The authors introduced it but did not mention that it has been appeared in China.
The authors should also mention the prevalence of other psychiatric diagnoses associated with the experience of the illness by COVID (clinically relevant depressive symptoms, anxiety, psychotic symptoms, etc).
Answer: We mentioned the initiation of COVID-19 in China.
But, we cannot find the appropriate place for the contents on the prevalence of other psychiatric disorders. We added this to the discussion section.
Methods. I have no comments. Psychometric instruments and all outcomes are well described.
Answer: I appreciate your comments.
Discussion. The authors should compare the prevalence of other psychiatric diagnoses in COVID-19 patients with the prevalence of PTSD.
Answer: We added the prevalence of other psychiatric disorder to the discussion section.
Round 2
Reviewer 1 Report
Thank you for reworking the article. I feel that it is a lot better now.
Author Response
Thank you for reworking the article. I feel that it is a lot better now.
Answer: We appreciate your valuable comment. Thank you.
This manuscript is a resubmission of an earlier submission. The following is a list of the peer review reports and author responses from that submission.